# Recent Advances in Acoustic Technology in Food Processing

**DOI:** 10.3390/foods12183365

**Published:** 2023-09-07

**Authors:** Daiva Zadeike, Rimgaile Degutyte

**Affiliations:** Department of Food Science and Technology, Faculty of Chemical Technology, Kaunas University of Technology, 50254 Kaunas, Lithuania; rimgaile.degutyte@ktu.lt

**Keywords:** ultrasound, safety monitoring, bioprocess regulation, biopolymer modification, extraction effectiveness, biocomposite development, quality assessment

## Abstract

The development of food industry technologies and increasing the sustainability and effectiveness of processing comprise some of the relevant objectives of EU policy. Furthermore, advances in the development of innovative non-thermal technologies can meet consumers’ demand for high-quality, safe, nutritious, and minimally processed foods. Acoustic technology is characterized as environmentally friendly and is considered an alternative method due to its sustainability and economic efficiency. This technology provides advantages such as the intensification of processes, increasing the efficiency of processes and eliminating inefficient ones, improving product quality, maintaining the product’s texture, organoleptic properties, and nutritional value, and ensuring the microbiological safety of the product. This review summarizes some important applications of acoustic technology in food processing, from monitoring the safety of raw materials and products, intensifying bioprocesses, increasing the effectiveness of the extraction of valuable food components, modifying food polymers’ texture and technological properties, to developing biodegradable biopolymer-based composites and materials for food packaging, along with the advantages and challenges of this technology.

## 1. Introduction

In recent years, EU and global efforts have been aimed at solving the problems of the efficient use of resources, ecology, and the development of food production technologies, to increase the sustainability and efficiency of processing. Various traditional processing and preservation methods, such as extrusion, filtration, extraction, drying, frying, cooking, fermentation, etc., are still widely used to process raw food materials. The underlying principle of most traditional food-processing methods depends on the use of high-temperature regimes to inhibit foodborne pathogens, thus ensuring the safety of food products [1]. Therefore, non-thermal technologies for food processing, such as ultrasound, irradiation, pulsed electric fields, cold plasma, and high hydrostatic pressure, have been widely evaluated [2]. These technologies enable prolonging the shelf life of the food, maintaining its nutritional, texture, and sensory characteristics, as well as increasing the bioavailability of food nutrients [3]. 

Acoustic technology is one of the sustainable alternatives to thermal processing, which is recognized as economically efficient and environmentally friendly and can be applied in the food (cereals, milk, meat, fruits/vegetables, drinks, etc.) production industry and also in the development of bio-based and biodegradable materials for safe food packaging [4,5]. This technology provides advantages, such as the intensification of technological processes, increased extraction efficiency, the modification of food components, maintaining its texture, organoleptic properties, and nutritional value, and ensuring the microbiological safety of the product [6,7,8,9,10,11]. Depending on the intensity, ultrasound (US) can be used for the activation or deactivation of enzymes, mixing, homogenization, emulsification, preservation, stabilization, ripening, and solid–liquid extraction to improve the extraction yields of active ingredients from different matrices. Moreover, the advances in the development of innovative non-thermal technologies can meet consumer demand for high-quality, safe, nutritious, and minimally processed foods [12]. 

In the food industry, most applications of high-power low-frequency US (>1 W/cm^2^; <100 kHz) are based on systems in which a liquid or a gaseous medium (such as air) is used for the propagation of the ultrasonic waves [13]. This type of US, known as power US, induces mechanical, physical, chemical, and biochemical changes through acoustic cavitation, caused by the formation, growth, and collapse of bubbles, releasing a large amount of energy. The energy is used in food-processing operations, such as drying, extraction, emulsification, and inactivation of pathogenic bacteria and their enzymes in the food matrix or on its surface [14,15]. 

Recent reviews on advanced ultrasound applications in various industrial operations have analyzed the basic principles of US generation to improve the efficiency of food production processes, resulting in shorter processing times and lower energy consumption [16,17,18], the ultrasonic degradation of food biopolymers (starch, proteins, etc.) [19], the green extraction of natural food components (e.g., polyunsaturated fatty acids, tocopherols, carotenoids, phospholipids, antioxidants, coloring compounds, essential oils, and seed oils) [20,21], as well as innovations related to the use of acoustic energy in food processing (i.e., low-frequency and high-intensity US) compared to thermal processing technologies (microwave technology, supercritical fluid extraction, pasteurization, sterilization, aseptic packaging, etc.) [4,22,23]. In fact, used alone or in combination with other methods, this technology can ensure high process yields and have positive impacts on the quality of foods [22]. 

Therefore, this review summarizes some additional aspects of the application of acoustic technology in the food production chain, starting from monitoring the safety and quality of raw materials and products, intensifying the bioprocesses, increasing the effectiveness of the extraction, modifying the food texture and technological properties, providing quality control, developing and assessing the quality of biopolymer-based composite materials, along with the advantages and challenges of acoustic technology. 

## 2. Ultrasound Generation Systems

Acoustic technology is based on mechanical waves, and the frequency, amplitude, and wavelength are intrinsic characteristics of acoustic waves (Figure 1). 

The frequency of US waves represents the number of cycles completed per second. The amplitude represents the strength (peak pressure), and the period represents the length of time to complete one cycle (Figure 1). The distance of a complete cycle is the wavelength. Low-frequency ultrasound leads to a longer cavitation bubble formation and growth period and bigger bubbles with more-powerful implosions, thus improving the refining efficiency, while high-frequency ultrasound leads to a shorter cavitation bubble growth period and the formation of smaller bubbles; thus, the cavitation effect weakens with increasing ultrasonic frequency. 

Thus, these characteristics impact the US waves’ ability to promote different effects on the sonicated medium. In this case, acoustic technology is classified by the frequency into high-frequency (1–100 MHz) and low-frequency (16–200 kHz) US [24]. High-frequency ultrasound (HFUS) generally has been applied in medicine for imaging diagnostics [25]. HFUS has shorter wavelengths and, for this reason, is absorbed more easily and is, therefore, less penetrating. This explains its use on superficial structures and, hence, its increasing application in the field of diagnostic medicine [26]. 

On the other hand, the application of low-frequency ultrasound (LFUS) promotes physical and chemical modifications on matrices such as food, polymers, alloys, and others [27,28]. 

In this case, the US-generation technique is a versatile system, which can be used in different operational applications depending on its configuration and the intensity of the generated acoustic energy [29]. The generation of US using ultrasonic bath or probe-based systems (Figure 2) takes place due to transducers with different geometries and diameters, which can transfer the acoustic energy directly or indirectly [4,21].

The types of acoustic systems (probe or bath) and the processing conditions, such as the frequency, power, duration, temperature, and sample-to-water ratio, influence the performance of ultrasonication. During the acoustic cavitation process, the formation of microbubbles occurs, causing certain physical and chemical modifications of the ultrasonically treated material [29]. The interaction of the cavitation-induced microbubbles and the food matrix can directly or indirectly lead to the formation of cracks and the initiation of pore formation in the physical structure of the food or food components [30]. For example, when applying US to the extraction process, the expected results highly depend on the chosen parameters, such as the applied power and frequency; however, US power is minimized in food manufacturing [20]. Meanwhile, the frequencies used in these processes are usually in the range between 20 and 100 kHz. When the frequency is increased, cavitation becomes more difficult to induce, as the period becomes too short for the cavitation bubbles to form and grow; therefore, higher US intensities would be necessary [20].

In solid–liquid systems, i.e., in the presence of solid particles, cavitation-induced bubbles collapse, causing the liquid to rapidly move from the bubble toward the solid’s surface, modifying the surface’s structure [20].

This process leads to a rapid transfer of heat and mass at the solid’s surface, modifying the surface’s structure. Cavitation energy is the energy absorbed by the microbubbles and is responsible for acoustic cavitation. In turn, the acoustic cavitation releases this energy through mechanical, thermal, and chemical modifications of the sonicated medium. In this way, methodologies based on physical properties, which allow the direct or indirect measurement of the applied acoustic energy, are widely used [22]. These effects have been observed in food matrices subjected to ultrasound-assisted extraction, homogenization, modification, stabilization, and emulsification [30,31,32,33,34].

## 3. Application of Acoustic Technology in Food Processing for Quality and Safety Control

### 3.1. Fruits and Vegetables

The extension of the shelf life of fresh or minimally processed fruits and vegetables is a key problem to be solved during their post-harvest storage [35]. Fully removing microorganisms on the surface of products remains a challenge in the food industry [35]. In the review of Chen et al. [36], US processing emerged as a novel tool for food preservation, providing antimicrobial effects due to the cavitation process. As water is used in this technology, it could be a promising method, which could be implemented in the washing step to obtain safe fresh or fresh-cut fruits and vegetables [36]. Hereby, US-assisted fruit drying reduces wastewater toxicity and energy consumption and improves productivity. 

US was reported to be effective for application in minimally processed products, causing only minimal losses in natural aromas and colors, maintaining an acceptable quality of fresh and cut fruits and vegetables and inhibiting or limiting the formation of microbes [37]. US (frequencies usually used during ultrasonic-assisted drying are between 20 and 40 kHz to avoid the continuous loss of sound wave energy) can be used as a pre-treatment prior to drying fruits and vegetables, since it increases the drying kinetics [38]. 

Ultrasound-assisted treatment is a pesticide-removal technology, which is safer for the environment and more effective at pesticide removal for a number of fruits and vegetables, including grapes, cabbage, carrots, tomatoes, and cucumbers [39,40]. For this purpose, in the study of Cengiz et al. [41], two kinds of US treatments, an ultrasonic bath at 40 kHz and an ultrasonic probe at 24 kHz, in combination with a low-intensity electrical current (1400 mA + 40 kHz, 800 mA + 24 kHz, and 1400 mA + 24 kHz) were tested for the determination of US’s effectiveness in the reduction of some important pesticide residues in tomato samples. These combinations led to a reduction of the captan, thiamethoxam, and metalaxyl residues by 94.24%, 69.80%, and 95.06%, respectively [41]. Lozowicka and co-authors demonstrated that, for strawberries, a 5 min cleaning step with US (40 kHz; 5 min) efficiently reduced sixteen pesticide residues by 91.2% [42]. Similarly, for lettuce surfaces, US with frequencies of 20–60 kHz can be used to remove insecticide contents by up to 89–95% after 8 min of US treatment without any changes in the nutritional properties [43]. It was found that the residue levels of organophosphorus pesticides (trichlorfon, dimethoate, dichlorvos, fenitrothion, and chlorpyrifos) on raw cucumber were significantly reduced by up to around 85% after US treatment (US bath, 40 kHz) for 20 min [44]. Concerning transient cavitation, under high-intensity ultrasound, bubbles attain the required size promptly and rupture, which would cause a high pressure (up to 100 MPa) and temperature (up to 5000 K) in a short period [36,44]. This causes the pyrolysis of pesticides and cell disruption, which promote reactions between the reactive species and the pesticide molecules. 

Acoustic technology has gained much attention due to its inhibitory effect on browning enzymes thanks to the capability of breaking the cell membranes. In particular, it has been discovered that US in combination with temperature and high pressure is more effective against polyphenol oxidase (PO) [45]. The PO in the original juice of various fruits was less inactivated than the US-treated purified form [46]. Ultrasonication at an intensity >200 W induced the aggregation and dissociation of PO particles and significantly decreased the α-helix structure. Similarly, pineapple juice had its content of PO reduced after 10 min of US treatment, as well as a viscosity decline of 75% [47]. PO decreased also on fresh cut potatoes after 5 min of US treatment without a change in color, while a 10 min treatment damaged the cells of potatoes [48]. The anti-browning effectivity, measured as PO activity increase, improved when ultrasound (40 kHz) was combined with ascorbic acid (1%) in fresh-cut apple in storage at 10 °C during 12 days [49].

In summary, US may be valid for the fresh-cut fruit/vegetable industry since the majority of components and sensory attributes are not affected. Furthermore, ultrasound can be considered as an alternative technique to heat treatments. 

### 3.2. Cereals and Cereal Products

Contamination of certain foods with toxins, produced by some organisms, poses challenges to ensuring food safety and quality. Cereal-based food products comprise vital nutrients that have the potential to be naturally contaminated by various fungi, as well as their secondary toxic metabolites, considered as mycotoxins, which can infect the crops before or after harvest and can be typically found in foods, such as cereals, dried fruits, nuts, and spices [50]. Cereal crops give filamentous fungi, such as *Fusarium* and *Aspergillus* spp., an important opportunity to grow using the edible parts of the plants, such as starch granules [51]. The result of the fungal invasion is grains shriveling and becoming more porous (Figure 3). 

Among various approaches with the aim of eliminating or at least reducing the presence of mycotoxins in food, ultrasonication is one of the novel techniques being discussed, partially due to the avoidance of direct heat during the processing [53]. As was shown, US treatment under different conditions allowed a reduction in the amounts of selected mycotoxins from about 40% (aflatoxins B1, B2, G1, and G2) [54] to 96% (aflatoxin B1, deoxynivalenol (DON), zearalenone (ZEA), ochratoxin A (OTA)) [55]. The degradation of mycotoxins in aqueous solutions and maize was significantly affected by the US intensity (2.2–11 W/cm^2^) and sonication time range from 10 to 50 min, showing DON being more stable than AFB1, ZEA, and OTA [54]. According to Liu et al. [55], during the initial stage of US treatment (1.1–1.65 W/cm^2^; 8 min), power US can promote the dissolution of AFB1 and ZEA, which were subsequently dissolved in water and were partly oxidized by free radicals under the effect of power US. A possible mechanism of mycotoxin degradation was proposed by Liu et al. in another study [56], when 550 W power US (20 kHz; 6.6 W/cm^2^) with a 13 mm probe was applied for processing aqueous AFB1, which was degraded by 85.1% after 80 min of US exposure, significantly reducing its bioactivity and toxicity because of mycotoxin degradation by the free radicals generated during the acoustic treatment.

The dissolution of mycotoxins in water also facilitates their degradation. In this case, the presence of water plays a critical role, as the covalent bonds of the water molecules break upon US treatment, thus yielding highly reactive radicals, hydroxyl and hydrogen, together with the subsequent formation of hydrogen peroxide outside cavitation bubbles, which then attack the organic molecules of toxins and initiate their decomposition. 

Due to the adverse health effects caused by mycotoxins, the EU has been working for almost two decades on the harmonization of mycotoxin standards for foods based on established international regulations and methods of analysis and sampling. Because of the high cost of state-of-the-art invasive, relatively slow, multi-step, and expensive wet chemistry methods and the reluctance of industry to perform representative sampling for food safety purposes, the EU felt obliged to enforce rather strict measures for sampling (EC 401/2006). Taking into account the above-mentioned aspects, there is a growing interest in the application of new, faster methods for inline or online monitoring of the presence of mycotoxins in food.

Rapid methods, such as near-infrared (NIR) spectroscopy, appear to provide a new approach to monitoring the quality of agricultural products [57,58]. Yearly, the potential of using IR spectroscopy for the detection of mycotoxins, including deoxynivalenols (DON), ochratoxin, fumonisins, aflatoxins, and fungal contamination in cereals and cereal products, has also been demonstrated [59,60]. However, the technique is not particularly suitable for routine batch analysis because of the limited applications, while the slow scan speed and low sensitivity appear to be the main disadvantages of using IR instruments.

Acoustic waves generated by an acoustic spectrometer, working in a range of 4.95–35.70 kHz and measuring the amplitude of the acoustic signal penetrating through the tested sample (thickness of grain layer: 30 mm; diameter: 50 mm) have been proposed to be applied to determine the preliminary level of mycotoxin contamination based on changes in the microstructure of unaffected (wholesome) and affected (contaminated/scabby) grains and aflatoxin-inoculated corn kernels [52,61,62]. The speed and non-invasive character of an acoustic method make it suitable to be used to carry out inline high-throughput analysis, by testing product/matrix porosity, which is influenced by the structure (grain size, shape) and moisture content of contaminated grain matrix, preferably at the point of harvest or in the cereal-processing chain before milling and further processing [61].

Moreover, the efficiency of US application in the reduction of mycotoxins in contaminated wheat-based products can be explored by using a multi-step prevention system: acoustic screening of grains with the elimination of contaminated grain from the production chain and, in the second step, a detoxification approach (e.g., ultrasonication and fermentation) [63,64]. Trakselyte-Rupsiene and co-authors used fermentation with lactic acid bacteria (LAB) in combination with an acoustic screening method (acoustic spectrophotometer; frequency: 10–60 kHz; duration: 10 s; thickness of the grain layer: 50 mm; diameter: 80 mm) for the prevention of *Fusarium* spp. mycotoxins in wheat grain, as well as in fermented products [63]. The study suggested that the acoustic technique used could identify DON, as well as DON-3-*β*-d-glucoside (D3G) contamination in raw wheat and is a promising tool in the wheat-grain-processing chain, and also, fermentation, using appropriate LABs, can reduce DON and D3G content in the fermented product by up to 75% and 84%, respectively. All authors indicated that the US power density and intensity, the solid–liquid ratio, and the US treatment modes significantly affect the degradation rates of mycotoxins. Even more, in an interesting study by S.O. Kerhervea et al. [65], the development of the US technique was reported, intended for quality control by measuring the mechanical properties of noodle doughs. These noncontact measurements were applied online during dough processing at a pilot plant, giving the advantage of avoiding contamination arising from direct contact with the monitoring equipment. Furthermore, the authors also declared that this ultrasonic technique could be promising if monitoring changes in the properties of noodles caused by proteases that can be associated with biological contamination, mycotoxins, sometimes present in wheat [65]. 

### 3.3. Beverages

Currently, juice is one of the most-popular beverages in the food industry. However, juice usually suffers from the loss of important nutrients, freshness, and quality during the thermal processing necessary for food safety and quality preservation. Thermal treatment (>70–90 °C) denatures proteins, inactivating microorganisms and enzymes, which causes undesirable changes in the juice and shorten its shelf life. Incomplete inactivation of enzymes results in browning and cloudiness [36] and changes in the biochemical, physicochemical, organoleptic properties, and nutritional components (vitamins, phenolic, etc.) of juices [66]. For example, the hot water treatment of tomato juice at 90 °C for 90 s resulted in low lycopene retention (67%) and changes in color [67]. Unwanted color reduction (50%) and a decrease in ascorbic acid content (15–40%) occurred in strawberry juice after thermal processing (90 °C, 5 min) compared to a pulsed electric field (100 Hz, 500 µs) [68]. 

To maintain the desirable sensory characteristics of food, such as taste, flavor, texture, and overall acceptability, some acoustic-assisted processes have been developed. 

The study of Lagnika et al. [69], investigating the effects of US processing on the physicochemical and nutritional quality of pineapple juice, showed US treatment (500 W; 20 kHz; probe diameter of 10 mm; 15 min; <65 °C) had a significantly lower browning degree. US was effective at retaining the total phenolic content and delaying microbial growth in pineapple juice as compared to the thermally treated or untreated juice sample during 60 days of storage at room temperature. The study demonstrated that US combined with mild heat pasteurization treatment (65 °C; 15 min) could be able to effectively inactivate the microorganisms and pectin methylesterase in pineapple juice, whilst preserving a relatively high amount of phenols [69].

According to Shen et al., US treatment (26 kHz, 9 min) of green grape juice preserved its sensory attributes, providing the significant inactivation of microorganisms (<1 log CFU/mL) and the enrichment of bioactive compounds (up to 482.47–543.62 μg/mL) [70]. Similarly, temperature-controlled US treatment (55 °C; 10 min; intensity 75%) can promote better appearance and odor in apple juice, indicating the improvement of the sensory characteristics through microbiological stabilization [70]. Evidence of the advantage of US as a nonthermal approach in stabilizing the food matrix was demonstrated on cherry tomato [71]. According to the authors, applying of dual-frequency US (20/40 kHz) for 10 min to cherry tomato resulted in preserved quality parameters (inactivation of microorganisms around 2.12–3.10 log CFU/g) and even higher retention (31.64–33.09 mg GAE/100 g) of the total phenolic compounds [71].

HPUS has recently been approved as a highly promising technology that can be adopted for several purposes in the winemaking process for the treatment of crushed grapes. The effect of US used at different amplitudes (30–90%) for different periods of time (2–10 min) showed that an increase in the amplitude and sonication time did not affect the main polyphenol contents of red wines [72]. The application of a pilot-scale power ultrasound system (30 kHz, 2500 W, 8 W/cm^2^) to crushed grapes facilitated the extractability of compounds from grapes to the must-wine, increasing the total phenol (18% and 23%) and tannin concentrations (43% and 30%) in the wine from less-ripened grapes and in the wine from partially rotten grapes, respectively [73]. The application of HPUS to obtain value-added red wines, using short maceration times, led to producing wine of a higher color intensity and a higher total polyphenol and anthocyanin content from grapes macerated for 4 h [74]. In this case, the application of US treatment (30 kHz, 2500 W, 8 W/cm^2^) led to enhancing the extraction of volatile compounds in the must, especially the free terpene and norisoprenoid content (from 3.37 to 4.92 μg/L). However, US caused the degradation of some phenolic compounds and vitamins, changes in color, the loss of anthocyanin, and other adverse effects on the food characteristics [74]. 

### 3.4. Milk and Dairy Products

The most-common effects of acoustic cavitation on dairy products were recently reviewed by Carrillo-Lopez et al. [75] and Chavez-Martínez et al. [76]. The main impacts of US treatment were observed on the physicochemical, functional, and microbiological properties of milk and dairy products. In the study by Carrillo-Lopez et al., the application of 10 min of sonication (50% and 100%; 24 kHz; 400 W in continuous mode) of fresh raw milk led to increasing the yield of Panela cheese by 24.29%. Moreover, increasing the US time resulted in an evident yellow tone of the cheese [75]. The differences between low- and high-intensity US, as well as the advantages and disadvantages of each one in terms of the processing, quality, and preservation of milk and dairy products were reported in detail by Chávez-Martínez and co-authors [76]. As low-power US (LPUS) (<3 W/cm^2^) does not have a destructive effect on raw material components in the food industry, its applications are focused on as quality control tools for the monitoring of the microbial growth, enzymatic reactions, fermentation, and gelling processes of milk [77,78]. Low-frequency US (22 kHz; 60–120 W/L) treatment, used to reconstitute powdered milk, yields a higher nutritional final product with a higher content of bioactive compounds, such as exopolysaccharides, vitamin C content, and antioxidants, when applied prior to fermentation, which improves the growth of fermenting bacteria [79].

HPUS modifies the biological, physical, and chemical properties of materials through destructive tests, and it has been used during the production of various dairy products [80]. The rheological properties of dairy products developed by sonication during the fermentation (sonotrode of 20 kHz; 20 W; 10% protein; 43.5 °C; pH 5.8–5.1) of Greek yogurt could facilitate the subsequent stirring within the production process of this product type [81]. LPUS applied to monitor cheese maturation by measuring the US velocity with a couple of narrow-band US transducers (1 MHz) showed that the US velocity increased with the ripening time from 1630 to 1740 m/s, depending on the cheese texture [82]. 

The possibility of using US as a pre-treatment method to improve the nutritional attributes of cheese was demonstrated by Munira and co-authors [83]. The authors, evaluating the potential of US, comparing it to different milk processing techniques, such as microwaves (MWs) and high-pressure (HP), for the antioxidant and angiotensin-I converting enzyme (ACE)-inhibitory activity of cheddar cheese during ripening, indicated that the antioxidant activity and ACE-inhibitory potential of cheeses made from pre-treated milk significantly increased in the following order: US-II (41 J/g; 20 kHz; milk flow rate of 35 mL/min) > HP (400 MPa; 15 min; <40 °C) > US-I (23 J/g; 20 kHz; milk flow rate of 15 mL/min) > MW (86.5 J/g; 3 min; <40 °C) > untreated control. 

Otherwise, acoustic treatments may not only consequently improve, but can also decrease the quality characteristics of foods. Thermo-sonication as a method for milk pre-treatment before fermentation offers the possibility to obtain gels with rheological properties superior to those obtained from conventionally heated milk. However, HPUS applied during the fermentation process has not shown the most-desired results for this product. According to Nöbel et al. [84], HPUS (45 kHz; 17 kW/m^2^) applied during the fermentation of skim milk resulted in the formation of lumps and increased graininess, which are textural defects in yogurt. Lara-Castellanos et al. [85] showed a cheese prepared with 30% ultrasound-modified (20%, *w*/*v*; sonicator power of 50%, with pulses of 10 s for 30 min; temperature was controlled at 35 ± 5 °C) casein delayed the appearance of molds, but gave lower overall acceptability due to the changes in the microstructural, functional, and textural properties of the ultrasonicated casein. 

The application of low-intensity acoustic energy could have less impact on the quality characteristics of foods. Thereby, the use of acoustic energy can have mixed effects on the taste, aroma, appearance, freshness, and texture of food products. This effect is related to the US processing conditions and the properties of the processed foods. Since no studies evaluating the health effects of food produced by acoustic technology have been found, it would be appropriate to assess the toxicity of ultrasonicated food ingredients on consumer health.

### 3.5. Meat and Meat Products

Recent studies have reported the prospective application of high-intensity US on fresh meat [86], mostly in tenderizing, brining, cooking and fermentation, thawing, freezing, and bacterial inhibition. US treatment increases meat tenderness and shortens the period of aging, without any effect on other quality parameters [87]. This is attributed to the rupture of the myofibrillar structure of the protein, collagen macromolecules’ fragmentation, and protein migration, among others, which accelerate proteolysis. 

In the study of Caraveo et al. [88], the redness of ultrasound-treated meat (40 kHz; 11 W/cm^2^; 90 min) was lower after treatment than that of control meat, but no difference was observed after Day 8 of storage. US can significantly decrease coliform, mesophilic, and psychrophilic bacteria in the meat during storage; however, the original microbial loads increased constantly during refrigeration. It has also been reported that the quality parameters of food products, such as the color, nutritional substances, and texture, are closely correlated with the heat transfer rate during the freezing process [89]. Ultrasound treatment is able to generate smaller ice crystals by accelerating the heat transfer, thus retaining the original quality properties of the frozen food products [90]. Visy et al. evaluated the combined effect of US-induced acoustic cavitation (20 kHz frequency; 5.09 W/cm^2^ power density; intensity of 100 W) and microbubbles during the brining of pork loin (Longissimus dorsi) [91]. The US brining enhanced the NaCl diffusion into the meat compared to meat brined under static conditions and the formation of microscopic pores on the surface of meat myofibers [91]. Xu et al., evaluating the changes of US-assisted (power of 350 W; frequency of 40 kHz; 10 °C) thawing on lamb meat quality and differential metabolite profiles during refrigerated storage, found that ultrasound-assisted thawing improved the water-holding capacity and increased the color of the lamb during refrigerated storage [92]. Furthermore, ultrasound-assisted thawing also reduced the sulfhydryl content in the lamb and inhibited protein oxidation. Moreover, potential metabolites associated with amino acids, carbohydrates and their conjugates, and peptides could be identified after US-assisted thawing [92]. HPUS, by producing high-speed jets, increases the temperature of the thawing water, subsequently generating asymmetric bubble collapse, improving heat transfer, thereby shortening the thawing process [93]. Slightly different results were reported by Bao et al. [94], showing increased tenderness and overall acceptability of dry-cured yak meat, but negatively affecting the meat color, smell, and taste after the US treatment (20 kHz; 200–400 W).

Therefore, the optimization of the sonication time for different applications is inevitable. Meat products are usually non-homogeneous and extremely attenuating materials, which make it difficult for US waves to transmit through the material, due to the inability to penetrate the inner parts of the product and the absorption of US by the outer layers. Localized heating and overheating are common phenomena in ultrasonication [93]. The standardization of the HIUS-assisted freezing process and product variables is a major challenge to scale this technology for industrialization. 

Sonication can also be used as a suitable tool to produce marinated food products with a lesser amount of salt (sodium chloride) in comparison to presently available commercial marinades. Gómez-Salazar et al. [95] studied the effect of acid marination assisted by power ultrasound (40 kHz; 110 W) on the quality of rabbit meat. It was observed that the ultrasound-assisted marinating increased the NaCl uptake in rabbit meat in comparison with marinating without additional US treatment. Furthermore, the acoustic treatment also reduced the time required for salting and the coloring of raw meat, allowing producing a product with a uniform salt content [95]. Contreras-Lopez et al. [96] observed that the application of high-intensity acoustic energy increased the overall salt concentration in pork loins and retained the color and quality. Furthermore, in the study by Sanches et al. [97], US treatment led to a higher NaCl content in a shorter time during beef wet brining, reduced the denaturation temperature of myofibrillar proteins, and did not affect the lipid oxidation of the beef when compared to samples in static brine. 

## 4. Inactivation of Microorganisms and Enzymes

The possibility of the occurrence or growth of pathogenic microorganisms in food at various stages of processing is a potential danger to the consumer; thus, various preservation technologies are being developed [98]. Although preservation methods, such as pasteurization, sterilization, and aseptic packaging, have been effective in the inactivation of spoilage microorganisms and enzymes, by altering the secondary and tertiary structure of proteins and damaging microbial cell walls [99], there are identified disadvantages of thermal treatment methods, such as the loss of heat-sensitive nutrients, i.e., vitamins, aromatic compounds, color pigments [4]. Furthermore, the fact that thermal treatment methods require high water consumption makes them expensive technologies [4]. 

Ultrasound technology is one of several novel nonthermal processing technologies that have been explored as an alternative to traditional thermal methods to avoid high temperatures to inactivate pathogenic microorganisms and their enzymes as consumers demand safe, minimally processed, and high-nutritional-value foods [100,101]. Bacterial inactivation effectiveness depends on the morphology of the bacterial cells, which influences their resistance to sonication. In general, it is considered that cocci, such as *Staphylococcus aureus*, are more resistant than rod-shaped bacteria; this may be due to their size and surface area. Cocci are smaller with less surface area, rendering them more resistant to ultrasonication, while larger cells tend to be more sensitive than smaller ones [102].

The inactivation of microbial enzymes occurs due to their depolymerization, caused by the acoustic cavitation or the binding of free radicals, which destabilizes the enzymes due to the changes in their structures [103]. Moreover, these mechanical, thermal, and chemical processes induce stress on microorganisms, causing the destruction of microbial cells and the inactivation of key enzymes [104,105]. Various researchers have reported that US could be effective against various dairy-related food enzymes, including alkaline phosphatase, lactoperoxidase, and γ-glutamyl transpeptidase and with varying effectiveness against the respective enzyme [106]. Typically, the enzyme activity decreases as the enzyme concentration increases, but higher solid content (i.e., high protein and fat) can enhance enzyme inactivation. 

In dairy systems, ultrasound can homogenize protein aggregates and cause whey protein denaturation. Pegu and Arya depicted that shorter duration and intensity (at 200 W for 4 and 6 min) resulted in a 1–2% increase in ALP activity [107]; however, as time and intensity increased, the ALP activity decreased. Enzyme activity can rise due to increased mass transfer and impart substrate availability, making enzymes more readily available for reaction. The effectiveness of microbial inactivation depends on the treatment conditions, including the frequency, intensity, duration, temperature, and pressure. According to Li et al. [108], the combined use of a US bath (300–500 W) and water at 55 °C for 10 min reduced the colony diameter of *Rhizopus stolonifer* in sweet potatoes. 

Apart from the factors above, milk and dairy products exert a sonoprotective effect on bacteria, such as *Listeria monocytogenes*, *Escherichia coli*, and *Pseudomonas fluorescens* [109]. Complex systems, such as those created by the addition of inulin, whey, lactose, and other sugars, hinder the transfer of energy produced by the acoustic cavitation through the beverage, thus protecting the bacteria [110]. On the other hand, high-power ultrasound can be coupled with thermosonication, and the main advantage of this method is pasteurization at lower temperatures than those used in conventional processing. In this regard, Monteiro et al. [111] applied thermosonication processing (19 kHz; 400 W) at different energy densities (0.3–3.0 kJ/cm^3^) as a nonthermal alternative to high-temperature short-time pasteurization (HTST) (72 °C/15 s) to pasteurize chocolate-flavored milk. The results showed that, with the increase in energy density, a product with a higher flow, but with a lower consistency index was obtained. Compared with conventional pasteurization, the ultrasound decreased the size of the fat globules, while it denatured the proteins at the same time [111].

Scudino et al. reported a higher microbial inactivation and stability during storage of Minas frescal cheese produced with sonicated milk (20 kHz, 160, 400, or 600 W) compared with heat-treated milk [112]. In these studies, ultrasound was associated with temperature. The association of ultrasound and temperature has been recommended due to higher microbial inactivation and the possibility of achieving the processing temperature in a shorter time [112]. 

On the other hand, Bermúdez-Aguirre and Barbosa-Cánovas [113] evaluated the microbial counts in soft cheese produced with milk subjected to thermal treatment using the US technique (400 W; 24 kHz; 63 °C and 72 °C) for different times. The treatments with lower US intensities (63 °C for 10 min, 72 °C for 15 s or 1 min) showed high microbial loads (>4 log) at the storage end (23 days; 4 °C). However, the treatment with the highest intensity (63 °C for 30 min) obtained the best result, maintaining the microbial load in a desirable pattern (<4 log), similar to conventional thermal treatment. In addition, due to the physical forces generated by acoustic cavitation, the particle size is reduced, there is a greater distribution of milk fat globules, and the structures of the milk protein are modified, thus improving the physicochemical properties and providing the creation of new products [112].

The potential of the combined treatment of ultrasound and nanoemulsions of *Litsea cubeba* essential oil against a common foodborne pathogen, *Salmonella*, was reported in a recent study by Ruiying Su et al. [114]. The authors declared the combination of these techniques to be very effective for the bactericidal cleaning of cherry tomatoes with the potential to find application in industry to control bacterial contamination on fresh produce. Bai et al. [115] reported a very high antibiofilm activity of low-frequency ultrasound (20 kHz, 300 W, 5 min) against *Escherichia coli* O157:H7, including the prevention of secondary biofilm formation. However, applying ultrasonication alone was not fully effective against bacteria inside the biofilm. Additionally, the removal of biofilm when tested on fresh fruits and vegetables was less effective. This coincides with the conclusions presented in another study [116], suggesting that ultrasound treatment alone is not sufficiently effective at controlling *E. coli* O157:H7 in food, and research is usually provided while applying US in combination with other techniques, or a suitable food group should be selected.

Acoustics can display synergistic or antagonistic effects on bacteria, yeasts, and molds when combined with other types of decontamination methods, such as chemical and thermal approaches [117,118]. Although the reported effects of US in the food industry are promising, studies are still scarce on some types of products. Many effects are not understood, and the results are divergent among some studies due to the variables used in the processing or the characteristics of the food matrix, making comparisons difficult. More studies are needed to understand how HPUS changes the physicochemical characteristics of the product and how this is perceived by the consumer since some physical changes or chemical reactions can provide unwanted sensory characteristics. Therefore, it is necessary to balance microbial, enzymatic, and sensory effects before acoustic technology may be widely applied in the food production industry.

## 5. Intensification of Bioprocesses

US technology has recently received increasing attention as a new tool for enhancing various bioprocesses in the food industry. The modern fermentation industry is highly competitive and innovative and appreciates the possibilities of new technologies, improving the efficiency of the fermentation process and the quality of the products [119,120]. In addition, the food industry also needs new analytical tools to comprehensively monitor fermentation processes. New processing and monitoring technologies, including acoustic technology, have been intensively evaluated to increase the efficiency of food fermentation, improving the biological activity of microorganisms [17,121,122].

The use of US in fermentation for process shortening and improving the cell growth of fermenting microorganisms due to acoustic cavitation has been shown for lactic acid bacteria and yeast. According to Ojha et al. [123], low-intensity ultrasonication at the exponential metaphase in a frequency range of 18–30 kHz accelerates the growth of *Saccharomyces cerevisiae* with a resulting reduction in fermentation time and in a 33.3% increase in yeast biomass growth. Nguyen et al. [124] demonstrated the potential of low-frequency US (20 kHz, 7–30 min) in the stimulation of *Bifidobacterium*, resulting in effective lactose hydrolysis and lactic acid production in milk during 24 h of fermentation, depending on the probiotic strain. The lactose consumption of bifidobacteria increased up to 2.5–3 times, in comparison with that in untreated samples [124]. In other research, ultrasonication (30 kHz; 50 W) for up to 5 min increased the efficiency of the *Limosilactobacillus reuteri* fermentation process of Bakraei juice, increasing the levels of lactic acid, the antioxidant capacity, and the anti-inflammatory properties [125]. A significant increase in *S. cerevisiae* biomass yield was reached by using ultrasonication for 60 min (28 kHz; 140 W) [126]. Moreover, Al Daccache et al. [127] reported a significant increase in *Hanseniaspora* spp. yeast biomass growth and glucose consumption and, also, a significant decrease in the ethanol yield during US-assisted apple juice fermentation, while Huezo and co-authors [128] reported US having negative effects on *S. cerevisiae* performance and viability, reducing the glucose uptake and ethanol production rate. The results indicated that both direct (23–32 W/L) and indirect (1.4 W/L) ultrasonication showed an inhibitory effect and mass transfer limitations [128]. The study on the effect of low-intensity US (58–94 W/L) applied in the lag, log, and stationary growth phases of *Lactobacillus plantarum* in apple juice fermentation revealed a significant impact of sonication in the lag and log growth phases [129], indicating the promotion of microbial growth and the intensified biotransformation of organic acids. In all cases, the authors emphasized the importance of performing the US-assisted fermentation process under controlled conditions and optimizing the US parameters for each case.

Furthermore, the improvement of the enzymatic hydrolysis of sustainable biomass is necessary in order to lower the enzyme requirement and processing time. US can be effectively applied for the improvement of the hydrolysis process by degrading the lignocellulose structure and by eliminating the mass transfer resistance, which contributes to an increase in the bioproduct yields with reduced processing time and enzyme consumption [130]. Various physical and chemical effects enhancing the enzymatic reaction can be attained by varying the US parameters [131,132]. Therefore, ultrasonication can influence the activity of enzymes in the esterification processes if the input energy is not too high to produce the deactivation of the enzyme [133]. 

Applying a high US power can lead to cell destruction, being considered as a microbicide, while low-power US increases the growth of microbial cells [134]. Ultrasonication enhances the cell membrane’s permeability, allowing mass transfer. Since acoustic cavitation generates transient and irregular pores on the cell membrane, macromolecules (proteins, lipids, carbohydrates) and other non-permeable extracellular substances can quickly enter the cell membrane after temporary pore generation [134]. Ultrasonication also provides the easy transportation of small molecules in a liquid, releasing intracellular waste molecules quickly, changing the membrane potential, activating the calcium channel, and accelerating mass transfer in the cell membrane [134].

Nevertheless, the presence of US influences the enzyme activity and stability and depends on the sonication parameters, such as the applied power and frequency. Numerous scientists have observed that enzymatic reactions performed under mild US conditions lead to an increased enzyme activity [135]. In the study of Nadar and Rathod [136], the use of US at appropriate frequency and intensity levels allowed enhancing the enzyme activity due to favorable conformational changes in the protein molecules. When enzymes were irradiated by the optimum ultrasound frequency, the enzymes underwent a favorable conformational change, which resulted in the enhancement of the enzyme activity [136]. Research carried out on commercial cellulase activation indicated an increase in enzyme activity by 13% due to low-frequency acoustic treatment (20 kHz; 800 W) with a short processing time (170 s) [137]. Wang et al. [138] reported that a short time (10 min) of US exposure with low intensity (24 kHz) at 15 W increased the cellulase activity by 18.17%; however, longer sonication times decreased the activity. The authors suggested that even shorter times of application were capable of increasing the activity with a low-frequency ultrasound exposure resulting in gains of productivity. The positive effect was due to the ability of ultrasound to increase the surface area of the enzyme molecules. The ultrasound energy absorbed by the enzyme molecules varied with the different ultrasound wavelengths, and also, it affected the stability of the enzyme, which ultimately resulted in a change of the catalytic activity. However, the potential of those applications is still limited widely due to the lack of proper information about their operational and performance parameters.

## 6. Modification of Food Biopolymers

Biopolymers (i.e., starch, protein, polysaccharides, fiber) are important food system components because of their functional properties, such as water solubility, water-/oil-holding capacity, swelling, foaming, and emulsification capacity, viscosity, and gelling [139,140]. The foods, by means of biopolymer incorporation, can have a lower viscosity (e.g., plant drinks), while others are intended to be highly viscous or gelatinous (i.e., soups, desserts, sauces). The main properties of biopolymer molecules that affect food texture are their morphology, composition, and interaction with other molecules [141]. Food biopolymers are widely used in food and nutraceutical delivery systems, so it becomes very important to improve their physicochemical and functional characteristics in order to be effectively used in human nutrition and have positive health effects. Some of the main factors impacting these important properties are conventional and emerging processing technologies, which may involve different levels of thermal and physical treatments, such as shear and pressure and exposure to low- to high-voltage electricity current or radiation [142].

Among the emerging technologies, US processing is a very promising technology with a high penetration rate for food and non-food applications, which has a high potential to affect biopolymers and their functionality [139,143,144]. The depolymerization process caused by acoustic cavitation mainly involves two mechanisms, mechanical degradation of the biopolymer molecules derived from the collapsed bubbles and chemical degradation caused by the chemical reactions between biopolymer molecules and high-energy molecules [145,146]. 

### 6.1. Modification of Starches

More specifically, US used for starch modification exhibits various advantages in terms of quality and higher selectivity and less processing time and, therefore, is considered as a sustainable processing technique. This technique was employed both on native starch and on gelatinized starch [147]. Although starch is a natural polymer, having a series of advantages, such as being renewable, low cost, and widely used in the food, chemical, and textile industries, medicine, and other fields, due to the limitation of its structure, natural starch has many deficiencies, such as insolubility in cold water, easy aging, and hardly reacting with other materials [148]. Therefore, it is important to modify starch to provide a product with special technological properties for food processing at an industrial scale [149]. As Sujka [150] reported, ultrasound processing (20 kHz; 170 W; 20 °C; 30 min) affected the average diameter and pore size distribution in rice, corn, wheat, and potato starches. According to Ding et al. [151], the morphological characteristics of retrograded starch can be changed due to high-power US treatment (20 kHz; 100–600 W; 30 min), resulting in a more-compact block-shaped structure. Zhang et al. reported the textural and cooking peculiarities of noodles prepared with different amounts of US-pretreated gluten similar to commercial ones [152]. With an increase of the US frequency (from 28 to 80 kHz), the solubility, water-holding capacity, and oil-holding capacity of noodles significantly increased (153.24%), due to the significantly reduced particle size of gluten (from 197.93 nm (untreated) to 110.15 nm (sonicated)) [152]. 

US treatment (30 min) mainly disrupts the amorphous region of starch granules, retaining their shape and size, but the hydration properties and pasting characteristics can be enhanced due to increased granule surface porosity [30]. The use of modified starch combined with other polymers, such as gelatin, and essential oils with an antimicrobial effect for edible coating production helps to reduce the limitations of starch and can be a low-cost alternative with great potential to increase the shelf life of fruits and improve their postharvest quality [153].

### 6.2. Protein Modification

The low digestibility of most natural plant-derived proteins, as well as their sensitivity to pH and temperature in food processing and preservation significantly limit their application in the food industry. Thus, it is important to modify the foods’ protein structure and enhance their functional properties [154,155]. 

Among the main methods of protein modification including physical technologies, such as microwave [156], cold plasma [157], and pulsed electric field [158], chemical technologies, such as glycosylation [159], and also, biological technologies, such as enzymatic hydrolysis [160], high-intensity ultrasound has been widely used in the modification of food proteins in recent years. The physical effects caused by acoustic cavitation involve the folding and unfolding of protein molecules with significant effects on the interactions between protein and water and between proteins [161]. Thus, high-intensity US can effectively regulate the polymerization and depolymerization of protein molecules, which affects the solubility, emulsification, foaming, and gelling properties of proteins. It is now widely used for the improvement of the properties of plant-derived functional foods or food ingredients with its convenience and safety of operation and the low-cost production [162]. 

Modifications to the functional and biological properties of proteins of cowpea pulse by ultrasound (100–200 W; 5–20 min) increased the protein solubility (57.26–68.85%), water-holding capacity (3.06–3.68 g/g), foaming capacity and stability (70.64–83.74% and 30.76–60.01%), emulsion activity and stability (47.48–64.26% and 56.59–87.71%,), and in vitro protein digestibility (88.27–89.99%) [163]. After sonication, the hydrophobic protein groups were exposed and the proteins were partially denatured, which increased the functionality. 

### 6.3. Polysaccharide Modification

As mentioned earlier, US processing can change the molecular structure of biopolymers, which is closely related to their biological activities, such as antioxidant, prebiotic, and anticancer [145,164]. According to Xiao et al. [165], ultrasound treatment (20 kHz; 20–80 °C) influenced *Flammulina velutipes* polysaccharide (FVP) chain conformation, reduced the viscosity and gel strength, and increased its thermal stability. Importantly, after in vitro fermentation of US-treated FVP, higher contents of short-chain fatty acids, promoting the growth of *Bifidobacterium* and *Brautella* and inhibiting the growth of pathogens, were obtained. As reported Yu et al. [166], low-frequency ultrasound (20 kHz; 400–1200 W; 30 °C) effectively degraded *Porphyra yezoensis* polysaccharides, increasing their antiproliferative effects on cancer cells. High-frequency (850 kHz; 70 W) ultrasonication (30 min; 40 °C) applied for the functionalization of rice bran significantly improved the extraction yield of alkali-soluble hemicelluloses, as well as their antioxidant activity [166]. Hydrothermal treatment of cereal bran can result in the degradation and depolymerization of the hemicelluloses, and the remaining insoluble polysaccharides are more effectively extractable in alkali; moreover, the alkaline solvent can disrupt hydrogen and covalent bonds, lowering the hemicellulose content [167]. In the case of the ultrasonication effect, molecules with a high molecular weight and long-chain length are broken more than shorter and stiffer chains [168]. The antioxidant activity of the extracted hemicelluloses may has been improved due to disrupting the cell walls, reducing the particle size, and enhancing the extraction of bound phenols affected by the acoustic cavitation and cellulase hydrolysis [169].

In conclusion, the previous studies confirmed the potential of acoustic technology as a method to modify food biopolymers for the improvement of their physicochemical and bioactive properties. It must be emphasized that the efficiency of US processing depends on various factors, including the frequency, US power density, time, temperature, and the macromolecular structure of the biopolymers. In order to obtain the desired results, it is recommended to optimize the US parameters and use the best conditions. This would help to fully exploit the potential of US technology in creating new functional properties of biopolymers for various food, pharmaceutical, and other industrial applications.

## 7. Increasing the Effectiveness of the Extraction 

Traditional techniques (solvent extraction, distillation (steam, steam/water, water, cold pressing) used for the extraction of natural nutritious and bioactive components require longer extraction times, but with lower yields, large amounts of organic solvents, and low extraction efficiency. Acoustic technology has the potential to be applied for the improvement of the extraction efficiency due to the degradation of cell walls, leading to higher product yields with lowered solvent consumption [170]. The operating principles are associated with the influence of various operating conditions, including the US frequency, power, and duration, reactor designs, and kinetics applied for acoustic intensification.

In the study of Song et al. [171], the US-assisted extraction at optimized conditions (250 W; 54.7 °C; 42.8 min) used to recover crude polysaccharides from maize cob residues led to obtaining the optimum extraction rate of 0.56 %, which was more efficient than that of the simple water-extraction method. Furthermore, the authors suggested that, during the US-based extraction, the US power should not be too high; otherwise, it will easily cause the breakage of the sugar chains and lead to some loss of the biological activity of the polysaccharide [171]. 

In recent years, the application of plant-derived bioactive components has shown an incremental trend in food applications, such as dietary-fiber-rich or enriched with natural antioxidants meat products [172,173]. Because of the need to explore sustainable protein sources, US-assisted extraction (100 W and 200 W; 5–20 min) used for the isolation of protein from cowpea significantly increased the protein yield from 31.78% to 58.96% [164]. Karki et al. [174] reported the use of high-power sonication as one of the pre-treatment methods with an improved protein extraction yield of 40–46% from defatted soy meal. In the study of Byanju et al. [175], the HPUS treatment of defatted soy flakes resulted in higher protein extraction yields when exposed to higher-power sonication compared to control soy flakes (90% and 68.7%, respectively), with lower (2.5 W/cm^3^) and higher (4.5 W/cm^3^) US power density. However, it was found that the sonication of the chickpea flours resulted in a decrease in the protein extraction yield, possibly due to high carbohydrate and fat, reducing the access to the proteins in the cell matrices [175].

Recently, acoustic-assisted extraction has been extensively employed for the extraction of bioactive compounds from fruits and vegetables and from food wastes, improving the yield, productivity, and selectivity and decreasing the extraction time [176]. US applied for d-limonene extraction from fresh sweet lime peel at 60 °C and optimized parameters (25 kHz; 80 W; *m*/*w* ratio 1:10) allowed obtaining the highest extraction yield (32.9 mg/g, 97%) and a 12-fold shortening of the extraction time compared to the solvent extraction method [176]. Furthermore, the US-assisted extraction of the total phenolics (TP) from fresh Moringa oleifera leaves, using the ultrasonic bath at the optimum conditions (extraction time of 26 min; temperature of 59 °C), resulted in the maximum TP content of 34.36 mg GAE/g d.w. and an antioxidant activity of 491.9 µmol TE/g d.w. in the extracts [177]. The extraction of anthocyanins from purple yam [178], employing an ultrasonic homogenizer at 750W in pulse mode at a low temperature and shorter time period (30 °C; 10 min), resulted in a higher anthocyanin content than the conventional method. Furthermore, in the work of da Rocha and Noreña, the maximum extraction yield of anthocyanins (45%) was reached after US-assisted extraction at 250–450 W for 5 min [179]. The maximum extraction yields of phenolics and anthocyanins from jabuticaba peels were observed after exposure for 10 min at 25 kHz, using an ultrasonic water bath [179]. Ultrasound-assisted extraction (20 kHz; 90 min in 0.2 N HCl) was reported to be an effective method to extract high ester-pectin and low-methoxyl pectin from the peels of red dragon fruits, whereas the yield was even twice (2.71%) that compared with the extraction without US pre-treatment [180].

In the case of the advantages of US-based plant polysaccharide extraction, the low energy consumption, short time, high efficiency, mild heating temperature, and no destruction of active ingredients during US processing can be emphasized [181]. On the other hand, US can destroy not only the cell wall of plants, but the whole pant; secondly, vibration can promote the uniform release of polysaccharides from cells and disperse them in solvents for better dissolution and extraction [182]. 

Alternatively, ultrasonic enzyme-assisted extraction is also an emerging technology in the food industry since it has advantages. Firstly, combined US–enzyme extraction (extraction temperature of 46.8 °C; ultrasound time of 42.3 min; pH 4.28; ultrasound power of 311 W; enzymes: papain, pectinase, cellulase, and α-amylase at dosages of 50, 250, 200, 100 U/g, respectively) was reported to be an exceptionally efficient extraction method for polysaccharides, providing higher antioxidant and biological activities [183]. The enhancement in the extraction efficiency was attributed to the increase in collisions between the enzyme and substrate caused by ultrasonication [184].

All studies confirmed that maximum extraction yields of bioactive compounds can be obtained with optimized extraction conditions, such as ultrasound power, time, temperature, and solvent-to-solid ratio combinations, depending on the raw material [185].

For US-assisted extraction, low-frequency, high-intensity US (<100 kHz; >1 W/cm^2^) results in strong shear and mechanical forces, which are desirable for extraction; however, high-frequency sound waves produce a large number of reactive radicals [7]. A constant low frequency is preferred due to the formation of fewer cavitation bubbles with a comparatively greater diameter, yielding a larger cavitation effect as compared to the higher ultrasonic frequency [186]. The power applied for extraction depends on the raw material matrix and the component to be extracted, and it varies in the range from 20 to 700 W [187,188].

## 8. The Development of Biomaterials

Recently, US has also demonstrated remarkable potential for the development of various biomaterials for food applications [189]. High-intensity acoustic emission was reported as a suitable tool for the production of novel materials without high temperatures and pressures or long reaction times. 

The US-based production of different types of biomaterials, such as lipid and carbohydrate nanoparticles, protein microspheres, microgels, and biocomposites, is categorized based on the physical and/or chemical effects induced by the acoustic energy. Modified nanocellulose, lignin nanoparticles, and bio-polyesters are among the most-promising future candidates for nanocomposite-based packaging films with high barrier qualities [190].

The application of high-power acoustic cavitation to chemical reactions is utilized to produce nanostructured biomaterials [191]. US is suitable in material synthesis and has been coupled with other methods (such as microwave, supercritical CO_2_, high-pressure processing, enzymatic extraction, etc.) to achieve a synergistic effect that improves the overall process [192]. Acoustic cavitation is capable of mixing and heating the precursor to concentrated energy spots that are intense enough to trigger high-energy chemical reactions, thereby synthesizing nanomaterials without the need for high temperatures and pressure or long reaction times, which are usually required in conventional synthesis approaches [23]. This is due to the complex and wide range of processes induced by US, leading to nano-biomaterials with various structures and modifications tailor-made for drug delivery and diagnostics applications. 

The application of sonochemistry in polymer science has been an interesting topic of research in recent years [193]. US-assisted polymerization is a novel, clean, and green technology, which can be investigated further by coupling with thermochemistry or flow chemistry. The recent utilization of high-frequency ultrasound (>100 kHz) for polymer synthesis has evoked new interest in the use of sonochemistry in the field of polymer chemistry, especially in chain growth polymerization reactions. 

Otherwise, biocomposites, which consist of a bio-based polymer matrix and an organic and inorganic filler with at least one nanoscale material, have the potential to be used as active packaging materials due to their enhanced mechanical, thermal, barrier, antimicrobial, and antioxidant properties [194,195]. In the study of Gholizadeh et al. [196], acoustic emission (optimum operating range between 100 and 750 kHz; sensor placed at a distance of 80 mm from the initial crack tip) was applied in the determination of the effect of toughening composite laminates by polycaprolactone nanofibers on matrix-cracking, fiber-matrix-debonding, and fiber-breakage mechanisms. Ameur et al. [197] investigated the damage mechanisms in carbon–flax hybrid composites during tensile tests monitored by the acoustic emission technique, consisting of two resonant piezoelectric sensors (PCA MICRO-80) with a frequency bandwidth of 100 kHz–1 MHz and a resonance peak of 300 kHz. 

Natural fibers are increasingly used as strengthening materials for the production of low-cost and low-weight biocomposites (other advantages include their non-abrasive nature, high specific properties, and also, biodegradability). Natural fiber is abundant and more affordable in comparison with synthetic fiber, specifically having lower density and energy requirements, renewability, no skin irritation, a higher strength-to-weight ratio, a higher aspect ratio length to diameter of around 100, and a higher strength and elasticity modulus, showing great potential as glass, carbon, or other synthetic fiber replacements [198]. However, limitations, such as the poor hydration properties and large scattering in the mechanical properties and, also, the insufficient understanding of the mechanisms controlling their mechanical behavior and failure modes, still limit the use of natural fiber-reinforced composites in non-structural applications [199]. Acoustic emission has proven to be a suitable tool for the evaluation of the entire volume of a material and its structure in real-time and for its high sensitivity to any process. Acoustic emission was successfully applied for single-fiber composite tests for the detection and localization of fiber breakage. A combination of digital imaging and acoustic emission led to characterizing the flexural behavior of flax biocomposites, improving the reliability of the damage investigation without limiting the failure mechanism to matrix cracking, interfacial failure, and fiber breakage, as expected in uncontrolled event monitoring [200]. 

The possibility of damage characterization using acoustic emission depends on the complexity of the fiber structure: the presence of multiple damaged zones can lead to the difficult interpretation of the acoustic emission results. The hydrophilic nature of natural fibers adversely influences adhesion to the hydrophobic matrix, resulting in low compatibility and strength; furthermore, a strong interfacial bond represents a key aspect of the durability of composites [200]. To overcome the weaknesses of natural fiber materials, the modification or pretreatment of natural fiber could be used to increase matrix bonding, reinforcement, and composite strength. Moreover, in order to overcome the low degradation temperature of natural fiber (~200 °C) before processing with thermoplastics with a temperature that is up to 200 °C, the interfacial treatments (surface treatment resins, additive, coating) need to be improved [198]. 

With respect to these issues, acoustic emission may constitute a useful technique for the monitoring of the mechanical behavior of natural fiber composites. 

## 9. Conclusions

Ultrasonication is considered as a relatively inexpensive and environmentally friendly technology compared to other physical and chemical food-processing technologies, developing a high potential for consumer acceptance. Acoustic technology has advanced the food-processing industry with its wide application in various processes, serving as a sustainable and low-cost alternative. As a result of these acoustic effects, US has been shown to be an excellent technique for the monitoring of cereals and cereal-based food safety, the promotion of microbial growth, and the intensification of bioprocesses, such as microbial fermentation. Acoustic treatment provides a good opportunity to inactivate microorganisms and enzymes when combined with heat and pressure. This triple combination serves a successful inactivation process at lower temperatures, which provides a solution for the industry to obtain fresh-like foods. Furthermore, the mechanisms and kinetics of microbial and enzyme inactivation are still poorly understood and need to be investigated. 

Acoustic technology is effective at modifying and controlling the size and shape of food components and improving the extraction rate, leading to higher extraction yields and the retention of product quality characteristics (texture, nutrition value, organoleptic properties), and improving shelf life. High-intensity acoustic emission was effective in the enhanced synthesis of biomaterials with improved properties and performance and, also, in the development of novel materials for food packaging at low temperatures and short reaction times. More attention should be paid to novel food and nutraceutical delivery systems to obtain more data on the effects of acoustic energy on the quality of food and non-food matrices. 

As all processing technologies have advantages and disadvantages, the adoption of one of them in the food industry should be thoroughly considered in order to optimize all the involved parameters. Further research is needed to enable the commercial realization of acoustic technology in food processing and biomaterials’ development, especially the alternatives for new applications.

## Figures and Tables

**Figure 1 foods-12-03365-f001:**
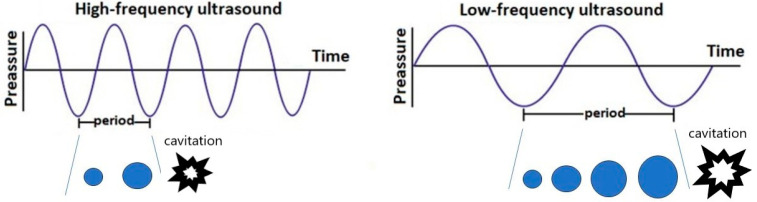
Acoustic waves of high- and low-frequency ultrasound. Blue dots—cavitation bubble.

**Figure 2 foods-12-03365-f002:**
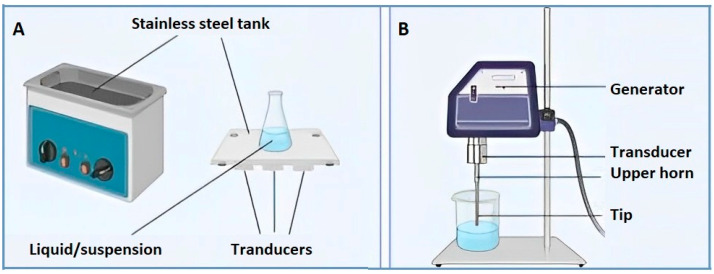
Acoustic systems: ultrasonic bath (**A**) and ultrasonic probe (**B**).

**Figure 3 foods-12-03365-f003:**
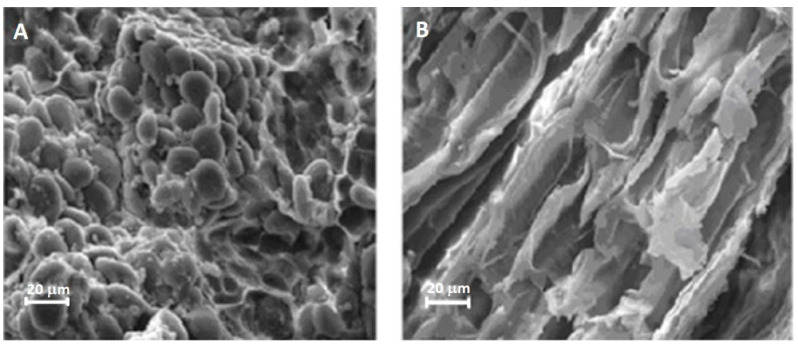
Healthy wheat kernels (**A**) with intact starch granules and kernels damaged by fungi (**B**) with stripped starch layers. Adapted from Juodeikiene et al. [52].

## Data Availability

Data is contained within the article.

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
