# Peer review of "Recent Advances in Acoustic Technology in Food Processing"

_foods, 2023, doi:10.3390/foods12183365_

Round 1

Reviewer 1 Report

Comments and Suggestions for Authors

This paper is a brief review on the potential uses of acoustic technology in food processing. The topic is of importance to the readers of the Journal. The technical merit is moderate. The coverage of the main issues are relatively limited. I believe that the authors need to generate a more comprehensive review dealing with potential implications in all food product categories. Otherwise the title is misleading. The language usage can be enhanced, especially for Section 1. Based on these and following points, I would like to recommend major revisions.

- Please explain Figure 1 in detail. 

- This review only contains text other than Figure 1. I believe that more compiled data, photos, figures, tables etc. has to be incorporated. 

- Section 3.1: Other than cereals, I believe that more food product categories have to be covered. Also I think that 3.1 and 3.3 are related. 

L234-239: I think that batch-to-batch variation is a huge issue here. Please  comment accordingly. 

-L192: Please do not use the plural form here or elsewhere. 

-L293: How do you define "plant drinks"?

- L315-319: Paraphrase to enhance clarity. 

This review has to be organized better, requires a more comprehensive approach. The authors should be able to present newcomers to the field an overall view of what has happened and will be happening in the field and mention real life industrial applications. If these issues are not covered, there is no point of publishing a new review for this topic. I do not believe that at this time this review is either focused or targeted enough.  

Comments on the Quality of English Language

The language usage can be enhanced, especially for Section 1. 

Author Response

Please find the response bellow:

This paper is a brief review on the potential uses of acoustic technology in food processing. The topic is of importance to the readers of the Journal. The coverage of the main issues are relatively limited.

I believe that the authors need to generate a more comprehensive review dealing with potential implications in all food product categories. Otherwise the title is misleading.

The language usage can be enhanced, especially for Section 1.

Based on these and following points, I would like to recommend major revisions.

Please explain Figure 1 in detail. Response: explanation for fig. 1 was added

This review only contains text other than Figure 1. I believe that more compiled data, photos, figures, tables etc. has to be incorporated. 

Section 3.1: Other than cereals, I believe that more food product categories have to be covered. Also I think that 3.1 and 3.3 are related. 

L234-239: I think that batch-to-batch variation is a huge issue here. Please  comment accordingly. 

-L192: Please do not use the plural form here or elsewhere. 

-L293: How do you define "plant drinks"?

- L315-319: Paraphrase to enhance clarity. 

 This review has to be organized better, requires a more comprehensive approach. The authors should be able to present newcomers to the field an overall view of what has happened and will be happening in the field and mention real life industrial applications. If these issues are not covered, there is no point of publishing a new review for this topic. I do not believe that at this time this review is either focused or targeted enough.  

The language usage can be enhanced, especially for Section 1. 

Response: The review was supplemented by additional information, some parts of review were rewritten. The language revised and improved. Please find the revised version.

Reviewer 2 Report

Comments and Suggestions for Authors

As part of the review, a synthesis of relevant and current literature for the application of ultrasound technology in food processing was conducted, along with the benefits and challenges of this innovative non-thermal technology.

The study is fully justified and motivated as it fills some gaps in terms of providing synthesized data on the application of acoustic technology in the food production chain, for improving food quality, for modifying food biopolymers, for increasing extraction efficiency and for developing biomaterials.

The introduction is well considered, accurately reflecting the current state of knowledge. The information included in this analysis is presented in a clear and well-structured manner. However, some parts are covered in depth, while others could be improved to make them more robust.

For example, section 3 (Application of ultrasound in food production chain) can be improved by providing more applications of UV treatment for this purpose. This part can be completed with Ultrasound Applications for Juice Production, when ultrasound technology is involved to improve juice yield and extract valuable compounds in juice production. Another direction may be the application of ultrasound treatment in red wine vinification, especially on crushed grapes to favour the extraction of their compounds. The application of ultrasound treatment in other food technologies can also be considered.

Section 6 (Increasing extraction efficiency) could also be completed with more applications.

The references are relevant to the topic, most of them published within the last 10 years.

Author Response

The authors thank the Reviewer for a very positive opinion.  The manuscript was supplemented by additional explanations and additional information. Some parts were rewritten. Please find the revised version of the Review.

Reviewer 3 Report

Comments and Suggestions for Authors

Comments to the author

The article provides a comprehensive overview of the applications and benefits of acoustic technology in various aspects of the food production chain, such as safety, bioprocesses, extraction, texture, and biopolymer-based materials.

However, the novelty of the article is questionable, as there are many articles that have been published on the same topic or using similar methods. These articles cover similar topics and issues as the article under review, such as the principles, mechanisms, effects, and applications of acoustic technology in food processing. Some of these articles also provide more detailed and comprehensive analyses of the current state-of-the-art and future directions of acoustic technology in food processing.

·         Rahman, MM, Lamsal, BP. Ultrasound-assisted extraction and modification of plant-based proteins: Impact on physicochemical, functional, and nutritional properties. Compr Rev Food Sci Food Saf. 2021; 20: 1457–1480. https://doi.org/10.1111/1541-4337.12709

·         Bonto, A. P., Tiozon Jr, R. N., Sreenivasulu, N., & Camacho, D. H. (2021). Impact of ultrasonic treatment on rice starch and grain functional properties: A review. Ultrasonics Sonochemistry, 71, 105383.

Therefore, the article should provide a more explicit and rigorous justification for its novelty claim. The authors should compare and contrast their work with existing literature and identify any gaps or limitations that their work addresses. The authors should also explain how their work provides a new perspective or insight on the research problem, how it offers a novel solution or application for a real-world issue, or how it generates new hypotheses or questions for future research. The authors should also include some figures or tables to illustrate their examples or results and show how they differ from or improve upon previous studies.

Please see the attached file for detailed comments.

Author Response

The authors thank for your opinion. Suggestions and recommendations helped to improve the Review. I, as guest editor of the special issue, have written this review journal editor's requests as a brief introduction to what will be covered in this special issue. The article does not pretend to be a broad specific overview of specific studies in one area.

The review was rewritten, by supplementing with additional information. Please find the revised version.

The manuscript was supplemented by additional explanations and additional information. Some parts were rewritten. Please find the revised version of the Review.

Reviewer 4 Report

Comments and Suggestions for Authors

Dear Authors,

detailed notes on the manuscript below:

1) the work would gain scientific value if the review of non-thermal physical methods additionally included, for example, a reference to methods based on the action of an electromagnetic wave (UV-C, magnetic field, electric field, neutron radiation, etc. - in relation to raw material and semi-finished products). On the basis of the above research, show the advantages of methods based on a mechanical wave,

2) I also suggest that individual methods related to ultrasonics (their effect) should be summarized in a table, specify the author of the method, working parameters used, type of product subjected to sound, final effect of treatment, etc.,

3) in my opinion, work should have a clear path: raw material - semi-finished product - finished food

3) 8. Conclusions - rather "Summary"

Author Response

The authors thank you for your positive opinion. Suggestions and recommendations helped to improve the Review. As a guest editor of the special issue, I have written this review according to journal editor's request as a brief introduction to what will be covered in this special issue. The article does not pretend to be a broad overview of specific studies in one area.

The review was rewritten, by supplementing with additional information. Please find the revised version.

The manuscript was supplemented by additional explanations and additional information. Some parts were rewritten. Please find the revised version of the Review.

Round 2

Reviewer 3 Report

Comments and Suggestions for Authors

The manuscript has been sufficiently improved to warrant publication.

Reviewer 4 Report

Comments and Suggestions for Authors

Thanks to the Authors for the changes and improvements.